# Determination of the Level of Cardiovascular Risk in 172,282 Spanish Working Women

**DOI:** 10.3390/diagnostics13172734

**Published:** 2023-08-23

**Authors:** Ángel Arturo López-González, María Albaladejo Blanco, Cristina Vidal Ribas, Pilar Tomás-Gil, Pere Riutord Sbert, José Ignacio Ramírez-Manent

**Affiliations:** 1Faculty of Odontology, ADEMA University School, 07009 Palma, Spain; angarturo@gmail.com (Á.A.L.-G.); pereriutord@gmail.com (P.R.S.); 2IdisBa (Balearic Islands Health Research Institute), 07004 Palma, Spain; joseignacio.ramirez@ibsalut.es; 3Investigation Group ADEMA SALUD IUNICS, 07003 Palma, Spain; 4Balearic Islands Health Service, 07003 Palma, Spain; maria.albaladejo@ibsalut.es (M.A.B.); cris.3.v.r@gmail.com (C.V.R.); 5Department of Medicine, University of the Balearic Islands, 07120 Palma, Spain

**Keywords:** risk equation, cardiovascular risk factors, women, occupational health

## Abstract

Introduction, objectives: Although cardiovascular events have been traditionally associated mainly with men, some data reflect an increase in women, which may even exceed their male counterparts, constituting the leading cause of death in working women in Spain. The objective of this present study was to analyze the level of cardiovascular risk in Spanish working women by assessing the influence of age, type of work, and tobacco consumption. Material, methods: A descriptive cross-sectional study was carried out in 172,282 working women from different Spanish geographical areas and from different companies between January 2018 and June 2020. A range of variables and risk factors were assessed and various cardiovascular risk scales were used to analyze the data. Results: An increase in cardiovascular risk was observed in the least qualified work groups, mainly corresponding to blue-collar workers, when using the SCORE or REGICOR risk equation. The prevalence of altered values for all the parameters analyzed (overweight and obesity, hypertension, dyslipidemia, diabetes, fatty liver, hepatic fibrosis, atherogenic indexes, and cardiovascular risk scales) was higher among blue-collar women. Age was the only factor that influenced all the cardiovascular risk scales studied, increasing risk when comparing the group of women aged 50 years and older with the others. Conclusions: Aging and belonging to the blue-collar job category meant worse results in the cardiovascular risk scales and in all the parameters analyzed. This is in line with numerous studies that argue that age and zip code are more influential than genetic code.

## 1. Introduction

Cardiovascular disease (CVD) has mainly been considered a male disease [1]; however, some data reflect an increase in this pathology in women, which has reached and even surpassed the number in men, becoming the leading cause of mortality in Spain with 230.5 deaths per 100,000 inhabitants in 2019 [2]. In CVD, there are sex-dependent variations in pathophysiology [3], symptoms [4], presentation, efficacy of diagnostic tests, and response to drug treatments. Women have been shown to have less obstructive disease but more microvascular coronary dysfunction than men, and their mortality rate is also higher one year after an acute myocardial infarction [5]. Similarly, it is more difficult for physicians to identify a coronary event in women, as such, prevention and treatment of this pathology in women must begin with awareness of the problem and understanding of its characteristics [1,6,7].

In women, cardiovascular events usually begin about 10 years [8] later than in men due to the protection provided by estrogens during the fertile period; however, after menopause, the levels of these hormones decrease and it is then that women become more exposed. Sex differences and hormones determine the basis for the mechanisms that regulate cardiovascular health and disease [9]. Estrogen receptors are distributed throughout all body tissues but their concentration varies in different tissues; In addition, estrogen concentrations vary throughout life, with an increase at puberty and a significant decrease at menopause. The latter shows evidence of vascular alterations in women, since 75% suffer from hot flushes and night sweats that are related to vasomotor instability [10].

The risk factors that have been most associated with the appearance of a cardiovascular event are dyslipidemia, arterial hypertension, tobacco use, obesity, diabetes, insulin resistance, and stress [11,12]. While physical exercise, regular consumption of fruit and vegetables, and limited alcohol intake have shown a protective effect [13,14,15]. An increase in cardiovascular risk has also been observed in the least qualified work groups, mainly corresponding to blue-collar work, which is associated with a low educational level [13,16].

Women have a significant 25% increased risk of coronary heart disease due to smoking compared to men, and heavy smokers up to six times more [17]. Diabetic, obese, and hypertensive women have three times higher risk, while in those with dyslipidemia, it can be double [14]. In women with heart failure, despite blood pressure control equal to that of men, arterial stiffness has a greater long-term negative effect [18].

The Women’s Ischemia Syndrome Evaluation study found that women with non-obstructive lesions had higher mortality rates and that cardiometabolic alterations related to increased adiposity—such as dyslipidemia, hypertension, and type 2 diabetes—were predictors of mortality [19], while the results obtained in the Nurse’s Health Initiative study found a lower incidence of ischemic heart disease among women following a healthy lifestyle together with a lower BMI compared to women with high rates of adiposity, high cholesterol levels, and lack of regular physical activity [20]. Obese women have greater central arterial stiffness, which can over time impair the left ventricular diastolic function and facilitate the development of heart failure, worsening quality of life due to the greater fatigue, dyspnea, and decreased exercise capacity, and this is more accentuated in women in comparison to men [21,22].

Cardiovascular disease is the leading cause of death for women in the Western world. We consider that risk factors in women have been very little studied and that their prevention should be addressed from an early age, so it is very important to know their frequency in women and their relationship with work activity [7]. To try to reduce the impact of the burden of morbidity, disability, and death from CVD, preventive intervention should be carried out through the detection and control of cardiovascular risk factors [13,23].

Diagnosis and treatment of cardiovascular diseases in women have been made difficult by not properly assessing the symptoms present in them, the lack of knowledge of the pathophysiological mechanisms that occur, and treating women differently from the standards laid out for men. Large-scale studies on epidemiology, prevention, and therapeutic options for cardiovascular disease in women should be conducted and analyzed by sex [24,25]. This will help us identify the factors that can increase the risk in women of suffering a cardiovascular event [10,26].

The objectives of the present study were to analyze the level of cardiovascular risk in Spanish working women by assessing the influence of age, type of work, and tobacco consumption.

## 2. Methods

A descriptive cross-sectional study was carried out in 172,282 working women belonging to different autonomous communities in Spain (Balearic Islands, Andalusia, Canary Islands, Valencian Community, Catalonia, Madrid, Castilla La Mancha, Castilla León, Basque Country) and from different employment sectors, especially hospitality, construction, trade, health, public administration, transport, education, industry, and cleaning between January 2018 and June 2020. Workers were selected based on their attendance to periodic occupational medical examinations.

Inclusion criteria:Belonging to one of the participating companies;Agreeing to participate in the study;Not having suffered a serious CVD event in the past (myocardial infarction, cerebrovascular disease…).

Of the 173,583 women initially included in the study, 699 were excluded due to not having data from all the necessary variables to calculate the cardiovascular risk indicators; 389 had suffered CVD previously; and 213 did not give permission to participate in the study. The final number of workers included in the study was 172,282 women. See flow chart in Figure 1.

Anthropometric, clinical, and analytical measures were carried out by the healthcare professionals of the different occupational health units that participated in the study, after standardizing the measurement techniques.

The following parameters related to cardiovascular risk were included in the assessment:

Weight (in kilograms) and height (in cm) were determined with a height bar scale (model: SECA 700 with a capacity of 200 kg and 50 g divisions, to which was added a SECA 220 telescopic height bar with millimetric division and 60–200 cm intervals);Abdominal waist circumference (cm) was measured with a SECA model 200 tape measure. The individual was placed in a standing position, with their feet together and trunk erect, abdomen relaxed, and the upper extremities hanging on both sides of the body. The tape measure was then placed parallel to the ground at the midpoint between the last palpable rib and the iliac crest [27];Blood pressure was measured in the supine position with a calibrated OMRON M3 automatic sphygmomanometer after a 10-minute rest period. A suitable cuff with 12 × 33 cm chambers was selected, and if the arm circumference was >33 cm, cuffs with 15 × 40 cm chambers were used. The cuff was placed on the skin about 2–3 cm above the elbow flexure, and the patient was instructed not to move or speak during the measurement. In addition, she had been asked not to eat, smoke, drink alcohol, coffee, or tea for 1 h before the visit. Three determinations were made at one-minute intervals, obtaining the mean of the three. Hypertension was considered when the values were greater than or equal to 140 mm Hg systolic or 90 mm Hg diastolic blood pressure, or individuals were previously diagnosed with arterial hypertension or under antihypertensive treatment;Blood glucose, total cholesterol, and triglycerides were determined by peripheral venipuncture after fasting for at least 12 h. Automated enzymatic methods were used. HDL was determined by precipitation with dextran sulfate Cl2Mg. LDL was calculated using Friedewald’s formula (provided triglycerides were less than 400 mg/dL). All the above values are expressed in mg/dL.
Friedewald’s formula: LDL = total cholesterol − HDL − triglycerides/5;

Blood glucose values were classified according to the recommendations of the American Diabetes Association [28], considering hyperglycemia > 125 mg/dL. Patients were classified as diabetic if they had a previous diagnosis or after obtaining a blood glucose level higher than 125 mg/dL, if they had an HbA1c ≥ 6.5%, or if the person was receiving hypoglycemic treatment. Cholesterol > 239 mg/dL, LDL > 159 mg/dL, and triglycerides > 200 mg/dL were considered high.

Cut-off points for the atherogenic indexes were [29]:Cholesterol/HDL (considered as high values > 5 in men and >4.5 in women);LDL/HDL and Triglycerides/HDL (high values > 3).

Metabolic syndrome was determined using three models [30]:The NCEP ATP III (National Cholesterol Educational Program Adult Treatment Panel III) considers metabolic syndrome when three or more of the following factors are present: waist circumference > 88 cm in women and 102 cm in men; triglycerides > 150 mg/dL or specific treatment for this lipid disorder; blood pressure > 130/85 mm Hg; HDL < 40 mg/dL in women or <50 mg/dL in men or specific treatment is followed; and fasting blood glucose >100 mg/dL or specific glycemic treatment;The International Diabetes Federation (IDF) model establishes as necessary the presence of central obesity, defined by a waist circumference > 80 cm in women and >94 cm in men, and at least two of the other factors mentioned above for ATP III;The JIS model uses the same criteria as the NCEP ATPIII but the waist cut-off points are those seen in the IDF model;A hypertriglyceridemic waist [30] required a waist circumference ≥ 94 cm in men, ≥80 cm in women, and triglycerides ≥ 150 mg/dL or treatment for hypertriglyceridemia.

The REGICOR scale is an adaptation of the Framingham scale to the Spanish population and estimates the risk of suffering a cardiovascular event over a 10-year period. It can be applied between 35 and 74 years of age. Moderate risk is considered >5% and high risk > 10% [31]. The SCORE scale is the version recommended for Spain and estimates the risk of suffering a fatal cerebrovascular event over a 10-year period. It is used between 40 and 65 years of age and risk is considered moderate with *numbers* 1–4, high ≥ 5%, and very high ≥ 10% [31].

To determine vascular age, calibrated tables [32] were used to assess the degree of aging of the arteries which can be calculated from the age of 30 years.

Vascular age with the Framingham model [33] considers age, sex, HDL-c, total cholesterol, systolic blood pressure, antihypertensive treatment, smoking, and diabetes. The scale can be calculated from the age of 30 years. Vascular age with the SCORE [33] model is calculated using age, sex, systolic blood pressure, smoking, and total cholesterol. As with the SCORE scale from which it derives, it can be calculated in people between 40 and 65 years of age. An interesting concept applicable to both vascular ages is avoidable lost life years (ALLY) [34], which can be defined as the difference between biological age (BI) and vascular age (VE).
ALLY = vascular age − biological age.

The different indicators were calculated using the following formulas:

Visceral adiposity index [30] (VAI):Men: VAI = (Waist/(39.68 + (1.88 × BMI)) × (Triglycerides/1.03) × (1.31/HDL)
Women: VAI = (Waist/(36.58 + (1.89 × BMI)) × (Triglycerides/0.81) × (1.52/HDL)

Waist triglyceride index [30]—Waist circumference (cm) x triglycerides (mmol).

Body shape index (ABSI) [30]:ABSI = Waist/(BMI^2/3^ × height^1/2^)

Normalized weight-adjusted index (NWAI) [30] was calculated as [(weight/10) − (10 × height) + 10], with weight measured in kg and height in m.

Conicity index [30]:CI = (Waist/0.109) × 1/√ weight/height

Lipid accumulation product [30]:In men—(waist circumference (cm) − 65) × (triglyceride concentration (mMol));In women—(waist circumference (cm) − 58) × (triglyceride concentration (mMol)).

Cardiometabolic index [30]:

Waist-to-height ratio × atherogenic index triglycerides/HDL-c.

Triglyceride glucose index [30] = LN (triglycerides [mg/dL] × glycemia [mg/dL]/2).

Triglyceride glucose index-BMI, Triglyceride glucose index-waist [30]:TyGindex-BMI = TyGindex × BMI
TyGindex-waist = TyGindex × waist

Atherogenic dyslipidemia is characterized by high triglyceride concentrations (>150 mg/dL), low HDL (<40 mg/dL in men and <50 mg/dL in women), and normal or slightly elevated LDL. If LDL levels are also high, we speak of the lipid triad [30].

Body mass index (BMI) was calculated by dividing weight by height in squared meters. Obesity was considered over 30. The waist-to-height ratio was considered risky over 0.50 [30].

Body Surface Index [29] (BSI) was calculated using the DuBois formula where w represents weight in kg and h represents height in cm.
BSA = weight^0.425^ × height^0.725^ × 0.0007184
BSI = weight/√BSA

A person was considered to have diabetes when the blood glucose levels were above 126 mg/dL (at least in two determinations) or they were under treatment for diabetes [35].

Formulas to estimate the percentage of body fat:

Relative fat mass [36]:76 − (20 × (height/p waist)) p waist = WC = Waist circumference
where height and waist circumference are expressed in meters. The cut-off point for obesity is 33.9% in women;
CUN BAE [37] (University of Navarra Body Adiposity Estimator Clinic) use the following formula:
−44.988 + (0.503 × age) + (10.689 × sex) + (3.172 × BMI) − (0.026 × BMI^2^) + (0.181 × BMI × sex) − (0.02 × BMI × age) − (0.005 × BMI^2^ × sex) + (0.00021 × BMI^2^ × age)
where male is 0 and female 1. The CUN BAE cut-off point for obesity is 35% in women;

ECORE-BF (Equation COrdoba Estimator Body Fat) [36]:

−97.102 + 0.123 (age) + 11.9 (gender) + 35.959 (LnBMI) Male = 0 Female = 1;

Palafolls formula [36]:

Men = ([BMI/waist] × 10) + BMI. Women = ([BMI/waist] × 10) + BMI + 10;

Deurenberg formula [36]:

1.2 × (BMI) + 0.23 × (age) – 10.8 × (gender) – 5.4 Male = 0 Female = 1;

Body Roundness Index [29] BRI:BRI = 364.2 − 365.5 × √{1 − [(waist/(2π))^2^ /(0.5 × height)^2^]}.
where WC represents the waist circumference.

Non-alcoholic fatty liver:Fatty liver index (FLI) [30]:
FLI = (e^0.953 × loge (triglycerides) + 0.139 × BMI + 0.718 × loge (ggt) + 0.053 × waist circumference − 15.745^)/(1 + e^0.953 × loge^
^(triglycerides) + 0.139 × BMI + 0.718 × loge (ggt) + 0.053 × waist circumference − 15.745^) × 100.

FLI scores of 60 and above indicate high risk.

Hepatic steatosis index (HSI) [30]:

HSI = 8 × ALT/AST + BMI (+ 2 if type 2 diabetes yes, + 2 if female);

Zhejian University index (ZJU) [30]:

BMI + FPG mmol L + TG mmol L+ 3 ALT/AST + 2 if female;

Fatty liver disease index (FLD) [30]:

BMI + TG + 3 × (ALT/AST) + 2 × Hyperglycemia (presence= 1; absence = 0)

Values < 28.0 or >37.0 excluded the possibility of NAFLD;

Bard scoring system (BSS) [37]:

Cut off for high-risk 38.
BMI ≥ 28 = 1 point, AST/ALT ≥0.8 = 2 points, type 2 diabetes mellitus = 1 point.

Cut off for high-risk 2 points.

A smoker was considered to be any person who had regularly consumed at least 1 cigarette/day (or the equivalent in other types of consumption) in the previous month or had quit smoking less than one year before.

Social class was determined from the 2011 National Classification of Occupations (CNO-11) and based on the proposal made by the social determinants group of the Spanish Society of Epidemiology [38]. We opted for classification into two categories: White-collar —directors/managers, university professionals, athletes, artists, intermediate occupations, and self-employed workers without employees. Blue-collar—unskilled workers.

### 2.1. Statistical Analysis

A descriptive analysis of the categorical variables was carried out, calculating the frequency and distribution of responses for each of them. For quantitative variables, the mean and standard deviation were calculated, while for qualitative variables the percentage was calculated. A bivariate association analysis was performed using the χ^2^ test (with a correction with the Fisher’s exact statistical test, when conditions required so) and Student’s *t*-test for independent samples. For the multivariate analysis, binary logistic regression was used with the Wald method, with the calculation of the odds ratio and the Hosmer–Lemeshow goodness-of-fit test performed. Statistical analysis was performed with the SPSS 27.0 program, and a *p* value of <0.05 was considered statistically significant.

### 2.2. Considerations and Ethical Aspects

This study was approved by the Clinical Research Ethics Committee of the Balearic Islands Health Area in November 2020, which was obtained with the following indicator IB 4383/20. The research team was always committed to following the ethical principles of health sciences research established at national and international level (Declaration of Helsinki), paying special attention to the anonymity of the participants and the confidentiality of the data collected. All patients signed written informed consent documents before participating in the study.

The identity of the participants will not be disclosed in any report of this study. The researchers will not disseminate any information that could identify them. In any case, the research team undertakes to strictly comply with Organic Law 3/2018 of December 5 on the protection of personal data and guarantee of digital rights, guaranteeing all participants in this study that they may exercise they rights of access, rectification, cancellation, and opposition of the data collected.

## 3. Results

The women in our study had an average age of less than 40 years and almost 60% were between 30 and 49 years old. Approximately one in three smoked and most belonged to the blue-collar group. Complete data on the characteristics of the sample are presented in Table 1.

Table 2 shows the average value of the different CVR scales calculated, according to the job position of the women. White collar-Blue collar.

All indicators of being overweight and having obesity (waist-to-weight ratio, BMI, NWAI, BRI, ABSI, VAI, conicity index, and formulas for estimating body fat) showed higher mean values in women belonging to the blue-collar group, and these differences were statistically significant. Something similar occurred with the scales to determine fatty liver (FLI, HIS, ZJU, FLD, LAP), hepatic fibrosis (BSS), and cardiovascular risk (SCORE scale, REGICOR scale, ALLY vascular age, Framingham, and SCORE). We also observed a similar trend with atherogenic indexes and other indicators related to cardiovascular risk (TyG, waist triglyceride index, and cardiometabolic index). The complete data are shown in Table 3.

The prevalence of altered values for all the parameters analyzed (overweight and obesity, hypertension, dyslipidemia, diabetes, fatty liver, hepatic fibrosis, atherogenic indexes, and cardiovascular risk scales) was higher among blue collar women. In all cases, the differences observed were statistically significant. All the data can be found in Table 3.

In the multivariate analysis, age was the only factor that influenced all the cardiovascular risk scales studied, increasing risk when comparing the group of women aged 50 years and older with the others. Those belonging to the blue-collar group also showed a negative influence on almost all the scales except total cholesterol, hypertriglyceridemic waist, lipid triad (where no influence was observed), and the REGICOR scale, where a slight protective effect was observed. Tobacco consumption was slightly protective against hypertension and the overweight and obesity scales but increased cardiovascular risk with the SCORE and REGICOR scales. All the data can be found in Table 4.

## 4. Discussion

Women are the most neglected in CVR programs, with few studies on the subject. In addition, menopause and post-menopause are times when the risk of cardiovascular disease (CVD) increases for women. This causes many doubts about the real management and evolution of women in these programs. Different studies have been carried out with estrogen treatments to reduce CVR in women; however, no benefits of this treatment have been observed to reduce the risk of cardiovascular events [39,40].

During this period, there is a redistribution of body fat with an increase in visceral fat, insulin resistance, and hyperlipidemia [41,42,43]. Hormonal changes act on the renin-angiotensin axis producing impaired vascular reactivity and endothelial dysfunction [44,45].

In 1999, the American Heart Association (AHA) published the first specific clinical recommendations for CVD prevention in women. Since this document, progress has been made in the awareness, treatment, and prevention of CVD in women [46]. It manifests itself ten years later in women than in men and brings with it a greater number of risk factors. In addition, women have a tendency not to identify symptoms in relation to CVD, as they are different from those in men, which causes a delay in diagnosis and greater risk [47].

However, cardiovascular events underwent a significant increase in the female population and has become the leading cause of death in working women [2]. Approaching menopause, estrogen levels in women decrease. We know that smoking, a sedentary lifestyle, excessive alcohol consumption, being overweight, and a diet rich in saturated fats all increase the risk of CVD, and that modifying lifestyle habits and changing these factors act preventively in CVD [48,49] but studies are based mainly on the male population [50,51].

Healthy lifestyle habits affect the risk of CVD in both sexes however, their influence varies between men and women. Smoking and excessive alcohol intake affect women more seriously. Estrogens have a protective effect on cardiovascular events, yet with aging these hormones decrease and, together with other elements such as obesity, sedentary lifestyle, or alcohol intake—which affect health and can even modify estrogen levels—thereby increase cardiovascular risk [47]. This motivated the carrying out of this study, whose main objective was to analyze risk by assessing the influence of age, type of work, and tobacco use.

One point to bear in mind is that for the detection and treatment of cardiovascular risk factors, sex differences between men and women identify CVD risk factors unique to women. We have learned that a single formula for stratifying cardiovascular risk is no longer valid for the entire population [52]. The 2011 American Heart Association (AHA) guideline for CVD prevention in women considered preeclampsia, gestational diabetes (GDM) [53], gestational hypertension, and systemic autoimmune disease [54] as disease-associated factors that increase the risk of CVD [46]; subsequently, the 2018 AHA/American College of Cardiology (ACC) multisocial cholesterol guideline and the 2019 ACC/AHA guideline on the primary prevention of CVD introduced the concept of “factors that increase risk” [55,56]. The risk increasing factors mentioned in these guidelines were premature menopause and pre-eclampsia [57].

We used the age of 50 as a cut-off point to distinguish pre-menopausal from post-menopausal women based on the SWAN study—which offered higher specificity and lower false positivity compared to definitions based on menstrual history—to examine an indicator of menopausal status [58]; however, taking age as an indicator of menopausal status can introduce a misclassification [59,60].

In our study, in line with other publications [10], age is the factor that most influences all the cardiovascular risk scales studied, also significantly increasing cardiovascular risk in all parameters studied. These results were obtained by comparing women aged 50 years or older with those in the group under 50 years of age, similar to other published studies [59].

When assessing smoking, we found an increase in cardiovascular risk both when evaluating the REGICOR scale and the SCORE scale, which coincides with other published papers [8,16,61]. Smoking is one of the main risk factors for CVD, and the leading cause of preventable death worldwide [62,63,64]; however, the odds ratio obtained was much higher when calculating the risk using the SCORE scale than against the REGICOR. This could form the basis for future research studies to assess which of the two formulas best detects cardiovascular risk in women from a given population.

In the multivariate analysis, when assessing the influence of smoking on the other risk factors, it appeared as a protective factor in women with hypertension and overweight/obesity. These results could surprise us or lead us to think that a mistake might have been made in the statistical treatment; however, we know that obesity is a very important risk factor for high blood pressure, and that these two pathologies often originate together [65,66]. Interestingly, tobacco and its components, especially nicotine, produce a decreased appetite and increased metabolism [67,68]. This can cause smokers to be closer to normal weight and therefore suffer less from obesity [69,70] and hypertension, with tobacco acting as a confounding factor.

Meanwhile, several studies have reported [20,71] that belonging to a less qualified labor group has a high prevalence of risk factors for cardiovascular diseases. This has been associated in different studies with a low quality of medical care and a limitation to high-quality medical resources [72]. This justification would not be applicable to our population, since the Spanish Health System is free and universal; however, the intake of certain foods of low nutritional quality, rich in fats, and processed foods favor the development of cardiovascular diseases. In 2008, the INTERHEART study found an increased risk of cardiovascular disease in people who ate a diet rich in meat, salty, and fried foods, and a lower risk of suffering from these pathologies in those who included fruit and vegetable intake in their diet [73].

This type of diet is highly influenced by socioeconomic level, in such a way that a low socioeconomic level consumes cheaper foods with a high caloric component, meat, and fat, along with a low consumption of fruits, vegetables, and wholemeal bread. This favors the development of obesity and an increase in other cardiovascular risk factors [74,75,76].

In our work, we obtained worse results in all the parameters studied in the blue-collar workers compared to the white-collar ones, with a statistically significant difference in all of them except for lipid parameters—where total cholesterol, lipid triad, and hypertriglyceridemic waist did not present significant values. In the same way, when assessing cardiovascular risk using the SCORE and REGICOR scales, both presented a significantly higher cardiovascular risk in women with manual jobs (Table 3).

Socioeconomic inequalities in health were identified and analyzed by the WHO in 1990. The population with the highest socioeconomic level was found to have lower mortality than groups with a lower socioeconomic level. These inequalities are presented according to different socioeconomic position indicators such as income, educational level, situation, and type of employment [49]; hence, according to the social level, those who enjoy a higher level have better health than those at lower levels. In Europe, mortality decreases more rapidly in people with a higher socioeconomic level than in those with lower economic resources, which further accentuates the increase in health differences according to socioeconomic class [77].

These differences in health due to socioeconomic status are maintained even in countries such as Sweden, which ranks ninth among the countries of the Organization for Economic Co-operation and Development (OECD) [78]. Sweden has created a comprehensive public health system with actions and protocols that seek to reduce social inequalities by including universal and accessible health care under equal conditions for the entire population. Socioeconomic status already influences the whole family from birth and adolescence, as shown in the Swedish study by Anderson et al. (2020) [79], in which parental occupational class—according to the Swedish Socioeconomic Classification Index (SEI)—was the most important factor in increasing cardiovascular risk factors in adolescents.

In Europe, women have to combine work with domestic work, spending much of their time doing housework. This is aggravated in low- and middle-income countries, where women spend more time in unpaid work than those residing in high-income countries, although there is also the influence of socioeconomic status within the country itself, which generates fewer professional prospects [80,81,82].

Higher mortality from all causes and particularly from cardiovascular disease in housewives has been described in several publications, a fact which according to these authors, could be explained by the “status syndrome”. This is a lack of control over one’s life and low social participation, which creates low self-esteem and chronic stress via dependence on the income of other family members. This situation can activate the inflammatory pathways and facilitate the development of cardiovascular diseases [83].

Stress has been related in developed countries to the type of work and responsibility in it. Many stressors are psychosocial in nature and depend on an individual’s response to them. Men and women are exposed to different demands and stressful situations in the work environment, even when they work in the same profession. In many cases, women work in lower-paid jobs, tend to stay longer in their jobs, and perform most of the housework, which further increases their stress by making it difficult to reconcile work and family life [84].

In relation to age, all the parameters evaluated worsened alongside aging in the same way, the different scales to measure cardiovascular risk increased inversely with the age of the woman. This coincides with studies published by other authors [85,86].

Finally, although we found worse results in all the parameters analyzed for blue-collar women, the most important difference appeared in the formulas to calculate metabolic syndrome and obesity, which could be influenced by visceral fat and the fact that all the parameters that are part of the metabolic syndrome were altered (high blood glucose and triglycerides, arterial hypertension, decreased HDL-C, and increased abdominal perimeter). This produced a significant increase in cardiovascular risk in blue-collar women compared to white-collar women, in agreement with previously published studies [87,88].

In 2021, Sonaglioni et al. [89,90] published two studies that we consider interesting to highlight. In these, they suggest that an increased Haller index (the ratio of chest transverse diameter over the distance between sternum and spine) due to a narrow antero-posterior chest diameter might be inversely associated with adverse cardiovascular outcomes in women. Further retrospective or prospective cohorts of women should be analyzed in their different chest wall conformation, to investigate if a concave-shaped chest wall and/or even a mild degree of pectus excavatum might be protective against adverse cardiovascular risk levels and/or outcome. In our work, as we do not have these measurements, we are unable to determine the cardiovascular risk according to the Haller index of the women included in our study.

We must not forget that while risk equations or tables are a valuable tool when making decisions regarding the prevention of cardiovascular events, a comprehensive clinical assessment of the patient must always be carried out.

These results provide us with an increase in knowledge about the most prevalent cardiovascular risk factors in women according to age, type of job (blue-collar/white-collar), and smoking, which can enable more correct actions on cardiovascular risk factors in women, and thereby reduce deaths and disabilities due to cardiovascular diseases [91].

### Strengths and Limitations

One of the strengths of this study is its large sample size, which makes it possible to obtain greater precision in the estimates and explore the presence of associations whose strength is weak. Another point is that the study population is made up of women workers from all areas of economic activity, which provides greater representation with respect to the general population. However, the fact that this study was carried out on workers who had attended their annual medical check-ups can be considered as a limitation, since it may incur the bias of the healthy volunteer. Additionally, the impossibility of calculating the Haller index with the parameters of our database is also a limitation in this work. Finally, the mean age was less than 40 years, which can also be a limitation when interpreting the results.

## 5. Conclusions

Age is the main risk factor for cardiovascular events while belonging to less qualified worker groups shows a negative influence on most of the factors analyzed to calculate risk. Likewise, tobacco, when applying the most widely used tables (SCORE and REGICOR), has a negative impact on the calculation of risk for cardiovascular events.

## Figures and Tables

**Figure 1 diagnostics-13-02734-f001:**
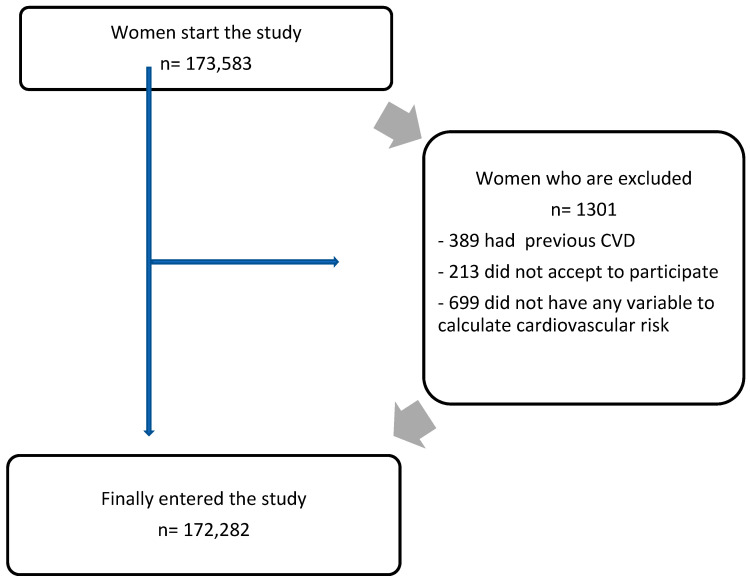
Participant flow chart.

**Table 1 diagnostics-13-02734-t001:** Characteristics of the women.

*n* = 172,282	Mean (SD)
Age (years)	39.6 (10.8)
Height (cm)	161.8 (6.5)
Weight (cm)	66.2 (14.0)
Waist (cm)	74.8 (10.6)
Systolic Blood Pressure (mmHg)	117.4 (15.7)
Diastolic Blood Pressure (mmHg)	72.6 (10.4)
Cholesterol (mg/dL)	190.6 (35.8)
HDL-c (mg/dL)	56.8 (8.7)
LDL-c (mg/dL)	116.1 (34.8)
Triglycerides (mg/dL)	89.1 (46.2)
Glycemia (mg/dL)	87.8 (15.1)
ALT (U/L)	20.2 (13.6)
AST (U/L)	18.2 (7.9)
GGT (U/L)	20.4 (19.7)
	**Percentage**
18–29 years	20.7
30–39 years	29.7
40–49 years	29.6
50–59 years	16.8
60–70 years	3.2
Blue-collar	69.8
White-collar	30.2
Non-Smokers	67.2
Smokers	32.8

HDL—Lipoprotein high density; LDL—Lipoprotein low density; ALT—alanine aminotransferase; AST—aspartate aminotransferase; GGT—gamma glutamyl transferase.

**Table 2 diagnostics-13-02734-t002:** Mean values of the different CVR scales according to work in women.

	Blue Collar	White Collar	Total	
	*n* = 120,212	*n* = 52,070	*n* = 172,282	
	Mean (SD)	Mean (SD)	Mean (SD)	*p*-Value
Age	39.7 (11.2)	39.2 (9.8)	39.6 (10.8)	<0.0001
Waist-to-height ratio (WtHR)	0.47 (0.06)	0.45 (0.06)	0.46 (0.06)	<0.0001
Body mass index	25.7 (5.3)	24.3 (4.7)	25.3 (5.2)	<0.0001
CUN BAE	35.7 (7.3)	33.9 (6.6)	35.2 (7.1)	<0.0001
ECORE-BF	35.7 (7.4)	33.8 (6.7)	35.2 (7.3)	<0.0001
Relative fat mass	32.4 (5.5)	31.1 (5.5)	32.0 (5.5)	<0.0001
Palafolls formula	39.1 (5.6)	37.6 (5.0)	38.7 (5.5)	<0.0001
Deurenberg formula	34.6 (7.3)	32.8 (6.4)	34.1 (7.1)	<0.0001
Body surface index	50.9 (8.3)	49.5 (7.5)	50.5 (8.1)	<0.0001
Normalized weight adjusted index	0.56 (1.40)	0.16 (1.26)	0.44 (1.37)	<0.0001
Body roundness index	2.8 (1.2)	2.6 (1.1)	2.8 (1.2)	<0.0001
Body shape index	0.069 (0.006)	0.069 (0.006)	0.069 (0.006)	<0.0001
Visceral adiposity index	2.8 (1.7)	2.6 (1.6)	2.7 (1.7)	<0.0001
Conicity index	1.1 (0.1)	1.1 (0.1)	1.1 (0.1)	<0.0001
Fatty liver index	19.3 (22.4)	15.2 (19.7)	18.2 (21.8)	<0.0001
Hepatic steatosis index	36.7 (7.0)	35.3 (6.4)	36.3 (6.9)	<0.0001
Zhejiang University index	37.3 (6.3)	35.9 (5.5)	36.9 (6.1)	<0.0001
Fatty Liver Disease index	30.4 (6.1)	29.1 (5.4)	30.0 (5.9)	<0.0001
Bard scoring system	0.67 (0.85)	0.51 (0.76)	0.63 (0.83)	<0.0001
Lipid accumulation product	18.7 (18.7)	16.5 (17.2)	18.1 (18.3)	<0.0001
Triglyceride glucose index	8.2 (0.5)	8.1 (0.5)	8.2 (0.5)	<0.0001
Triglyceride glucose index-BMI	221.2 (49.9)	198.4 (44.2)	207.3 (48.6)	<0.0001
Triglyceride glucose index-waist	616.4 (104.2)	600.9 (98.9)	611.7 (102.9)	<0.0001
Triglyceride glucose index-WtHR	3.8 (0.6)	3.7 (0.6)	3.8 (0.6)	<0.0001
Waist triglyceride index	78.0 (46.4)	73.0 (41.9)	76.5 (45.1)	<0.0001
ALLY vascular age SCORE *	4.5 (5.2)	3.4 (5.0)	4.2 (5.2)	<0.0001
SCORE scale *	0.54 (1.03)	0.32 (0.79)	0.47 (0.97)	<0.0001
ALLY vascular age Framingham **	1.8 (12.4)	-1.3 (10.6)	0.9 (11.9)	<0.0001
REGICOR scale ***	2.9 (2.2)	2.9 (2.2)	2.9 (2.2)	0.594
nº factors of metabolic syndrome NCEP ATPIII	0.9 (1.1)	0.7 (1.0)	0.9 (1.1)	<0.0001
nº factors of metabolic syndrome JIS	1.0 (1.2)	0.8 (1.0)	1.0 (1.1)	<0.0001
Cardiometabolic index	0.80 (0.55)	0.72 (0.49)	0.77 (0.53)	<0.0001
Atherogenic index total cholesterol/HDL-c	3.5 (0.9)	3.4 (0.9)	3.4 (0.9)	<0.0001
Atherogenic index triglycerides/HDL-c	1.7 (1.0)	1.5 (0.9)	1.6 (1.0)	<0.0001
Atherogenic index LDL-c/HDL-c	2.1 (0.8)	2.0 (0.8)	2.1 (0.8)	<0.0001

CUN BAE—Clinica Universitaria de Navarra Body Adiposity Estimator; ECORE-BF—Equation Cordoba Estimator Body Fat; ALLY—Avoidable lost life years; SCORE—Systematic Coronary Risk Evaluation; REGICOR—Registre Gironi del Corazón; NCEP ATP III—National Cholesterol Evaluation Program Adult Treatment Panel III; JIS—Joint Interim Statement; HDL—Lipoprotein high density; LDL—Lipoprotein low density. (*) *n* = 60,250 blue collar, *n* = 25,046 white collar, *n* = 85,296 total; (**) *n* = 94,185 blue collar, *n* = 42,436 white collar, *n* = 136,621 total (***); *n* = 92,174 blue collar, *n* = 40,489 white collar, *n* = 132,663 total.

**Table 3 diagnostics-13-02734-t003:** Prevalence of altered values of the different CVR scales according to work in women.

	Blue Collar	White Collar	Total	
	*n* = 120,212	*n* = 52,070	*n* = 172,282	
	Percentage	Percentage	Percentage	
Waist-to-height ratio > 0.50	23.6	16.7	21.5	<0.0001
Body mass index obesity	18.5	11.3	16.4	<0.0001
CUN BAE obesity	51.8	38.7	47.8	<0.0001
ECORE-BF obesity	50.8	37.8	46.9	<0.0001
Relative fat mass obesity	50.8	41.3	47.9	<0.0001
Palafolls formula obesity	75.7	66.6	72.9	<0.0001
Deurenberg formula obesity	70.9	62.9	68.5	<0.0001
Hypertension	14.9	10.4	13.5	<0.0001
Total cholesterol ≥ 200 mg/dL	37.7	35.2	36.9	<0.0001
LDL-c ≥ 130 mg/dL	33.0	29.6	32.0	<0.0001
Triglycerides ≥ 150 mg/dL	8.5	6.8	8.0	<0.0001
Glycemia 100–125 mg/dL	11.2	7.6	10.1	<0.0001
Glycemia ≥ 126 mg/dL	1.6	0.7	1.3	<0.0001
Metabolic syndrome NCEP ATPIII	10.9	6.5	9.6	<0.0001
Metabolic syndrome IDF	10.5	6.6	9.4	<0.0001
Metabolic syndrome JIS	12.6	7.7	11.1	<0.0001
Atherogenic dyslipidemia	4.3	3.3	4.0	<0.0001
Lipid triad	1.1	0.9	1.0	<0.0001
Hypertriglyceridemic waist	1.7	1.3	1.6	<0.0001
Atherogenic index total cholesterol/HDL-c moderate–high	12.2	9.9	11.5	<0.0001
Atherogenic index triglycerides/HDL-c high	7.6	5.9	7.1	<0.0001
Atherogenic index LDL-c/HDL-c high	13.7	11.2	13.0	<0.0001
SCORE scale moderate–high	5.5	3.1	4.8	<0.0001
REGICOR scale moderate–high	18.1	17.8	17.9	<0.0001
Fatty liver index high risk	8.5	5.7	7.7	<0.0001
Hepatic steatosis index high risk	47.4	37.9	44.7	<0.0001
Fatty liver disease index high	45.8	41.1	44.5	<0.0001
Bard scoring system high	15.8	10.5	14.3	<0.0001

CUN BAE—Clinica Universitaria de Navarra Body Adiposity Estimator; ECORE-BF—Equation Cordoba Estimator Body Fat; SCORE—Systematic Coronary Risk Evaluation; REGICOR—Registre Gironi del Corazón; NCEP ATP III—National Cholesterol Evaluation Program Adult Treatment Panel III; IDF—International Diabetes Federation; JIS—Joint Interim Statement; HDL—Lipoprotein high density; LDL—Lipoprotein low density.

**Table 4 diagnostics-13-02734-t004:** Logistic regression analysis.

	Age ≥ 50 Years	Blue Collar	Smokers
	OR (95% CI)	OR (95% CI)	OR (95% CI)
Hypertension	4.73 (4.60–4.88)	1.36 (1.31–1.40)	0.96 (0.93–0.99)
Total cholesterol ≥ 200 mg/dL	4.56 (4.45–4.68)	ns	ns
LDL-c ≥ 130 mg/dL	4.44 (4.33–4.55)	1.07 (1.04–1.09)	ns
Triglycerides > 150 mg/dL	2.41 (2.32–2.50)	1.19 (1.14–1.24)	ns
Glycemia ≥ 126 mg/dL	6.06 (5.76–6.37)	1.95 (1.82–2.08)	ns
Metabolic syndrome NCEP ATPIII	3.98 (3.85–4.12)	1.60 (1.54–1.67)	ns
Metabolic syndrome IDF	2.78 (2.69–2.88)	1.54 (1.48–1.61)	ns
Metabolic syndrome JIS	4.07 (3.95–4.20)	1.58 (1.52–1.64)	ns
AI Cholesterol/HDL-c moderate-high	4.52 (4.38–4.66)	1.13 (1.09–1.17)	ns
AI Triglyceride/HDL-c high	2.71 (2.61–2.82)	1.22 (1.17–1.28)	ns
AI LDL-c/HDL-c high	4.44 (4.31–4.57)	1.13 (1.09–1.17)	ns
Hypertriglyceridemic waist	5.02 (4.56–5.53)	ns	ns
Atherogenic dyslipidemia	3.09 (2.94–3.25)	1.18 (1.11–1.25)	ns
Lipid triad	5.02 (4.56–5.53)	ns	ns
Waist-to-height ratio > 0.50	1.40 (1.36–1.44)	1.52 (1.48–1.56)	0.97 (0.94–0.99)
BMI obesity	1.55 (1.50–1.59)	1.74 (1.69–1.79)	0.96 (0.94–0.99)
CUN BAE obesity	4.50 (4.38–4.63)	1.62 (1.59–1.66)	0.96 (0.94–0.98)
ECORE-BF	3.85 (3.75–3.95)	1.62 (1.59–1.66)	0.96 (0.94–0.98)
RFM obesity	1.53 (1.49–1.57)	1.44 (1.41–1.47)	0.95 (0.93–0.97)
Palafolls obesity	2.72 (2.63–2.81)	1.50 (1.47–1.54)	0.94 (0.92–0.96)
Deurenberg obesity	25.32 (23.61–27.15)	1.34 (1.31–1.37)	0.94 (0.92–0.96)
Diabesity	5.47 (5.07–5.90)	2.26 (2.51)	ns
Fatty liver index high risk	1.56 (1.50–1.63)	1.47 (1.40–1.55)	ns
Hepatic steatosis index high risk	1.92 (1.82–2.03)	1.42 (1.35–1.49)	ns
Zhejiang University index	1.96 (1.85–2.07)	1.55 (1.47–1.64)	ns
Fatty liver disease index high risk	1.67 (1.58–1.77)	1.18 (1.12–1.24)	ns
Lipid accumulation product high	1.73 (1.69–1.78)	1.32 (1.29–1.36)	ns
Bard scoring system high	14.26 (13.78–14.75)	1.36 (1.31–1.42)	ns
SCORE moderate–high	57.23 (28.39–88.56)	1.36 (1.25–1.49)	6.62 (6.16–7.12)
REGICOR moderate–high–very high	1.95 (1.89–2.01)	0.92 (0.89–0.95)	1.06 (1.03–1.09)

CUN BAE—Clinica Universitaria de Navarra Body Adiposity Estimator; ECORE-BF—Equation Cordoba Estimator Body Fat; SCORE—Systematic Coronary Risk Evaluation; REGICOR—Registre Gironi del Corazón. NCEP ATP III—National Cholesterol, Evaluation Program Adult Treatment Panel III; IDF—International Diabetes Federation; JIS—Joint Interim Statement; HDL—Lipoprotein high density; LDL—Lipoprotein low density; AI—atherogenic index; BMI—Body mass index.

## Data Availability

The study data is stored in a database that complies with all security measures at ADEMA-Escuela Universitaria. The Data Protection Delegate is Ángel Arturo López González.

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
