# Peer review of "Determination of the Level of Cardiovascular Risk in 172,282 Spanish Working Women"

_diagnostics, 2023, doi:10.3390/diagnostics13172734_

Round 1

Reviewer 1 Report

See my comments in attached word file.

No major remarks regarding the english language.

Author Response

Summary: Demographics and important cardiovascular risk factors in an employed Spanish population of women were recorded and compiled. From this data a high number of indexes and risk SCORES were calculated. Only a small share of the available women was excluded from analysis due to missing data or other reasons. The results from the present study confirm that ageing is linked to many cardiovascular risk factors, and moreover, that “blue-collar workers” and women with lower income have a less healthy lifestyle and little higher cardiovascular risk.

Article: The first aim of the study was to analyse the level of cardiovascular risk in Spanish working women. How do you know if you succeed in your ambition and how can “the level of cardiovascular risk” be presented clearly to the reader?

Since the present descriptive and cross-sectional population study only included women who are free from previous cardiovascular events, employed at selected companies in Spain, and only those who attend an offered occupational health check-up, it is possible that the selected population not completely mirror the true working women population in Spain. Nevertheless, one strength of this study is that data are collected from a large population.

We appreciate your comments and proceed to respond to your recommendations as well as modify them in the manuscript.

Method section: Details about the data collection process is however very vague “different Spanish geographical areas and from different companies”. How many companies were represented? Type of companies? Number of cities? Small villages included or only staff from large cities? Are all provinces in Spain represented? How large share of the staff in companies are expected to attend occupational medical check-up in Spain? Did the medical staff in the primary care or medical staff at the companies register all individual personal data on-line? Or did the researcher open medical records to search for data in all 173 000 subjects? Who were reviewing the database for correcting typing error?

Following their recommendations, we have added in our paper the Autonomous Communities of origin and the most frequent types of jobs in our population.

To make it easier to read and locate, we have highlighted the changes in the paper in red.

Review

In general, the introduction gives the reader a comprehensible overview of the topic, reviewing data from previous studies about different risk factors and incidence of cardiovascular events in women with adequate references added. The importance of implementing prevention strategies to decrease the number of events in women is mentioned but a paragraph with a historical perspective could be added to review how death rates in cardiovascular disease have changed during the last decades in the western world.

Methods: Data collection include basic parameters like demographic data, smoking history, office blood pressure and analysis of circulatory biomarkers. It seems that the same equipment and methods were used at all health care units for blood pressure and anthropometric measurements, and blood analysis was done with similar methods at all sites, which is excellent.

In my perspective, the method section present too many formulas. My suggestion is that you reduce the number of formulas. Focus more on your primary data, stick to limits for higher risk and the definition of different diagnosis.

Results: The table 2 show that regardless of which calculated parameter you look at, a significant difference in found between the two groups of workers. Few readers can interpret and understand the difference between all calculated parameters that are presented in the tables.

I suggest that you move part of the parameters from the result section to supplement material if you find them necessary to present.

Some titles in the reference list are in Spanish, some references were published without any referee control system.

Thank you very much for your observation, we have registered the references in English, except for those that the authors have only referenced in Spanish and that is how we have found an indication to cite them.

Specific comments:

Introduction:

First sentence, line 31. Who has always considered that cardiovascular disease to be a male disorder? Please add a reference here or erase the first sentence that stand alone in introduction text.

We have changed the word always to mainly, and we have added the reference.

Line 36. “Women have been shown to have less obstructive but more extensive coronary artery disease than men”. Which reference show this?

We have modified the sentence construction, and we have added the reference.

Line 68. “Obese women have greater arterial stiffness that affects diastolic function”. This can be expressed more clearly. Greater central arterial stiffness can by time impair the left ventricular diastolic function.

We appreciate your recommendation, and we have proceeded to change the phrase as indicated.

Methods

Line 188: “The ground at the height of the last floating rib”. Is this the standard method? Any reference?  

Thank you very much for your comment, the sentence was left incomplete, we have proceeded to complete the sentence and add a bibliography.

Line 120-123: Blood pressure method. Measured in random upper arm? Average value of right and left? Was cuff width matched to circumference of the individual arm?

As requested, we have added the following clarification to the paper.

A suitable cuff with 12 x 33 cm chambers was selected, and if the arm circumference was > 33 cm, cuffs with 15 x 40 cm chambers were used.

The cuff was placed on the skin about 2-3 cm above the elbow flexure, and the patient was instructed not to move or speak during the measurement. In addition, he had been asked not to eat, smoke, drink alcohol, coffee or tea for 1 hour before the visit

Line 203: “A person was considered to have diabetes when the blood glucose levels were above 126 mg/dl (at least in two determinations) or they were under treatment for diabetes or had a BMI of 30 kg/m2 or more[33].” Is BMI > 30 considered as diabetes without any positive blood glucose test?

Thank you very much for your warning, it is a writing error. We have deleted the last sentence.

Results:

Table 1 can be improved in layout.

I am sorry, but I do not know exactly what you are referring to.

Table 1-4 have headings but lack footnotes.

Done

Extensive number of calculated parameters are presented that all come from a limited number of variables.

We are very grateful for your comment, but one of the objectives of this study was to assess different cardiometabolic risk parameters by applying different scales for each one of them so that it could be seen that regardless of the scale applied, the risk existed, which is why we consider that applying so many scales can improve the study.

Similar parameters are found in table 3 and 4 with different categorisation of data.

You are indeed right, both tables show the same parameters, but table 3 assesses the prevalence of cardiometabolic risk parameters according to type of work, while table 4 performs a multivariate analysis using logistic regression and assesses how the risk of presenting these cardiometabolic risk parameters increases according to age, type of work and tobacco consumption.

Discussion:

The different text paragraph can be developed and adding references with contradicting findings could make the discussion more interesting.

Line 328. “Cardiovascular events may have undergone a significant increase in the female population” Are cardiovascular events more common nowadays then earlier in all countries? When did this rise in cardiovascular events occur?

Thank you very much for your observation. The full sentence reads “However, cardiovascular events may have undergone a significant increase in the female population until it became the leading cause of death in working women”.

While it is true that in recent years the number of cardiovascular events has decreased, both in men and women, in many countries. According to information from the Centers for Disease Control and Prevention, heart disease is the leading cause of death worldwide in women.

https://www.cdc.gov/healthequity/features/heartdisease/index.html

The same results are offered by the World Health Organization (WHO), in its latest published reports.

https://www.who.int/news-room/fact-sheets/detail/the-top-10-causes-of-death

The 2019 Global Burden of Disease study (Lancet)

https://www.thelancet.com/journals/lancet/article/PIIS0140-6736%2821%2900684-X/fulltext

Much of the discussion deals with monitoring of risk factors, impact of lifestyle and prevention strategies to decrease the incidence of cardiovascular disease. Only little space is used for interpretation of the results found in the present study and comparing this data with previous studies.

We have expanded the discussion with other studies of cardiovascular risk in women.

Dear reviewer, we have proceeded to answer all the questions raised and have made changes to the manuscript.

We trust that the answers will be to your liking and we appreciate all the recommendations made to improve our manuscript.

Sincerely

Reviewer 2 Report

The Article is well written and very interesting.

I congratulate the Authors for their findings.

The Authors listed all main determinants of cardiovascular risk in spanish working women.

I totally agree with their findings. However, among the anthropometrics, the Authors did not evaluate (measure) the women’s chest wall conformation.

Recent evidences suggest that an an increased Haller index (the ratio of chest transverse diameter over the distance between sternum and spine) due to a narrow antero-posterior chest diameter, might be inversely associated with adverse cardiovascular outcome in women population. Further retrospective or prospective cohorts of women population should be analyzed in their different chest wall conformation, to investigate if a concave-shaped chest wall and/or even a mild degree of pectus excavatum might be protective against adverse cardiovascular risk level and/or outcome. This aspect could be reported in the Limitations section. The Authors could cite the following references (PMID: 33961159 and PMID: 34485034).

Author Response

The Article is well written and very interesting.

I congratulate the Authors for their findings.

The Authors listed all main determinants of cardiovascular risk in spanish working women.

I totally agree with their findings. However, among the anthropometrics, the Authors did not evaluate (measure) the women’s chest wall conformation.

Recent evidences suggest that an an increased Haller index (the ratio of chest transverse diameter over the distance between sternum and spine) due to a narrow antero-posterior chest diameter, might be inversely associated with adverse cardiovascular outcome in women population. Further retrospective or prospective cohorts of women population should be analyzed in their different chest wall conformation, to investigate if a concave-shaped chest wall and/or even a mild degree of pectus excavatum might be protective against adverse cardiovascular risk level and/or outcome. This aspect could be reported in the Limitations section. The Authors could cite the following references (PMID: 33961159 and PMID: 34485034).

Dear reviewer, firstly thank you for your comments. We also appreciate the references you have provided us, as we were unaware of the existence of these articles.

After reading both articles, we found them of great interest, so we have expanded the discussion by referring to these works.

Following their recommendations, we have also included it in the limitations of our study.

To facilitate the location of the modifications in the paper, we have written them in red.

We appreciate your recommendations and trust that the modifications will be to your liking.

Sincerely

Round 2

Reviewer 1 Report

You have responded to most of my questions adeuquately. I have not reviewed all formulas in the paper but assume that this will be done later by others if manuscript is considered for publications. 
You have added a long paragraph about how chest configuration can be evaluated by implementing a formula that you never used in the discussion section. This red text paragraph should be shortened and revised.
See my few comments in the attached PDF version.

I have no major remarks.

Author Response

Review 1

Dear reviewer, once again thank you for your advice. We have proceeded to modify the last paragraph of our paper as you have recommended to us.

We trust that you will like the new wording, and we appreciate all the recommendations made to improve our manuscript.

Sincerely